# Antioxidant Activity and Molecular Docking Study of Volatile Constituents from Different Aromatic Lamiaceous Plants Cultivated in Madinah Monawara, Saudi Arabia

**DOI:** 10.3390/molecules26144145

**Published:** 2021-07-07

**Authors:** Amr Farouk, Mohamed Mohsen, Hatem Ali, Hamdy Shaaban, Najla Albaridi

**Affiliations:** 1Flavour and Aroma Chemistry Department, National Research Center, Cairo 12622, Egypt; hamdy_asn@yahoo.com; 2Madinah Region Municipality-Quality Agency-Food and Environment Laboratory, P.O. Box 4952, Al-Madina Al-Munawara 41412, Saudi Arabia; mohmedmohsen28@gmail.com; 3Food Technology Department, National Research Center, Cairo 12622, Egypt; haali@ksu.edu.sa; 4Food Science and Nutrition Department, College of Food Science and Agriculture, King Saud University, Riyadh 12372, Saudi Arabia; 5Department of Physical Sport Science, Nutrition and Food Science, Princess Nourah Bint Abdulrahman University, P.O. Box 84428, Riyadh 11671, Saudi Arabia; naalbaridi@pnu.edu.sa

**Keywords:** essential oil, GC-MS, *Mentha longifolia* L., *Mentha spicata* L., *Origanum majorana* L., antioxidant potential, oxidative stress, in silico studies

## Abstract

A comparative study of volatile constituents, antioxidant activity, and molecular docking was conducted between essential oils from *Mentha longifolia* L., *Mentha spicata* L., and *Origanum majorana* L., widely cultivated in Madinah. The investigation of volatile oils extracted by hydrodistillation was performed using Gas Chromatography-Mass Spectrometry (GC-MS). A total number of 29, 42, and 29 components were identified in *M. longifolia, M. spicata,* and *O. majorana* representing, respectively, 95.91, 94.62, and 98.42, of the total oils. Pulegone (38.42%), 1,8-cineole (15.60%), menthone (13.20%), and isopulegone (9.81%) were the dominant compounds in *M. longifolia* oil; carvone (35.14%), limonene (27.11%), germacrene D (4.73%), and β-caryophyllene (3.02%) were dominant in *M. spicata* oil; terpin-4-ol (42.47%), *trans*-sabinene hydrate (8.52%), γ-terpinene (7.90%), α-terpineol (7.38%), linalool (6.35%), α-terpinene (5.42%), and *cis*-sabinene hydrate (3.14%) were dominant in *O. majorana* oil. The antioxidant activity, assessed using DPPH free radical–scavenging and ABTS assays, was found to be the highest in *O. majorana* volatile oil, followed by *M. spicata* and *M. longifolia*, which is consistent with the differences in total phenolic content and volatile constituents identified in investigated oils. In the same context, molecular docking of the main identified volatiles on NADPH oxidase showed a higher binding affinity for *cis*-verbenyl acetate, followed by β-elemene and linalool, compared to the control (dextromethorphan). These results prove significant antioxidant abilities of the investigated oils, which may be considered for further analyses concerning the control of oxidative stress, as well as for their use as possible antioxidant agents in the pharmaceutical industry.

## 1. Introduction

Madinah Munawara is an important city in the Middle East with a relevant historical heritage and millions of visitors annually, located at the northwest of the Arab peninsula. Agriculture is one of the main activities in Madinah, with many crops, including medicinal plants [1]. Medicinal plants represent about 12% of the total floral species in the Arab peninsula and include 300 species belonging to 72 families. The *Lamiaceae*, also known as *Labiatae* or the mint family, is widely grown in the Arab peninsula, especially in Madinah. *Mentha longifolia* L. (Madany or Habak), *Mentha spicata* L. (Balady), and *Origanum majorana* L. (Doosh) are traditionally used alone or mixed with tea and other herbs as refreshment drinks and in cuisine dishes for their aroma and flavor. Moreover, such mints are used in folk medicine to treat nausea, bronchitis, anorexia, ulcerative colitis, liver diseases, and other symptoms [2]. The *Lamiaceae* family contains 236 genera, with *M. longifolia* and *M. spicata* belonging to the genus *Mentha* L. and *O. majorana* belonging to the genus *Origanum* L.

The biological activity of the essential oils of *M. longifolia* cultivated in Madinah has been studied by many authors and associated with antimicrobial, antimalarial, antileishmanial, antioxidant, antimutagenic, and other activities [3,4,5]. *M. longifolia* displayed a lower antimicrobial activity against *Leishmania donovani* promastigotes compared to other lamiaceous plants, like *Mentha australis* or *Teucrium polium* L., and also lower antifungal properties toward *Candida krusei* and *C. glabrata* [4]. In contrast, *M. longifolia* oil possesses stronger antioxidant and antibacterial activity compared to *M. polegium*, due to its higher flavonoid content, which makes it a good protector for cardiovascular and throat health [3,5]. The chemical composition of *M. longifolia* cultivated in Madinah is a debated issue because of remarkable discrepancies, raising the need for more investigation of the oil [4,6,7]. To the best of our knowledge, no reports have analyzed the biological activity of *M. spicata* or *O. majorana* cultivated in Madinah. However, the essential oil of *M. spicata* cultivated in Pakistan and Tunisia displayed effectively scavenged DPPH free radicals, inhibition of linoleic acid peroxidation, and high activity against *S. epidermidis* and *S. aureus*, as well as Gram-negative cells of *Salmonella* spp. and *E. coli* [5]. Similarly, *O. majorana* oil showed antioxidant, antimicrobial, cytotoxic, and acetylcholinesterase properties based on plants cultivated in Tunisia [8].

The antioxidant activity plays an important role in the regulation of redox homeostasis and oxidative stress reduction. Reactive oxygen species (ROS) generated during oxidation and rancidity of fats and oils are the main reason for oxidative damage, which causes cell or oxidative stress associated with many chronic diseases, like diabetes, cancer, and cardiovascular disorders. Therefore, antioxidants are used extensively to maintain the balance of redox homeostasis and reduced levels of oxidative stress [9]. Nowadays, molecular docking has been applied as a mechanistic tool to study the inhibition of enzymes that negatively affect antioxidant activity, like NADPH oxidase (NO), which is responsible for ROS generation [10].

The present study aimed to investigate chemical constituents of *M. longifolia* essential oil cultivated in Madinah to clarify the contradiction found in the literature, and also to unveil the volatile constituents of *M. spicata* and *O. majorana*. The antioxidant activity of the investigated oils was evaluated based on DPPH radical scavenging and β-carotene bleaching assays with the determination of the total phenolic content. The antioxidant potential of the main volatiles was evaluated in silico through molecular docking with NO enzyme, using dextromethorphan (DEX) as a positive control. The investigation of such *Lamiaceae* species opens the perspective towards a better understanding of their biological efficiency and its interpretation based on the chemical composition of the studied oils.

## 2. Results and Discussion

### 2.1. Chemical Composition of Volatile Oils

Hydrodistillation of the plant samples under investigation produced colorless to pale yellow oil, with 0.32 ± 0.05, 0.29 ± 0.04, and 0.26 ± 0.06% in Madinah essential oil and 0.42 ± 0.08% for M. longifolia, M. spicata, and O. majorana, respectively. The percentage yield was calculated on a dry weight basis. The chemical composition of volatile oils from aerial parts of M. longifolia, M. spicata, and O. majorana was characterized by GC-MS (Table 1, Figure 1A–C). A total number of 29, 42, and 29 components were identified, representing respectively 95.91, 94.62, and 98.42% of the total oils. Pulegone (38.42%), 1,8-cineole (15.60%), menthone (13.20%), and isopulegone (9.81%) were the dominant compounds in the M. longifolia oil, which is consistent with Ibrahim et al. [4], who studied the same species in Abyar Al-Mashy, Madinah, despite remarkable quantitative differences due to geographical variation (Table 1, Figure 1A). In contrast, entirely different profiles were revealed by Salman et al. [7] and Anwar et al. [6], the first study reporting menthone, eucalyptol, and isomenthone, while the second one reporting carvone, limonene, and dihydrocarveol as the major components. Our above findings are in line with previous reports on volatiles of the same species cultivated in Egypt [11], with quantitative differences, while the essential oils of M. longifolia cultivated in Tunisia and Tajikistan have an entirely different profile [12,13]. This is likely due to differences in environmental and geographic conditions.

The volatile oil extracted from M. spicata showed that carvone (35.14%), limonene (27.11%), germacrene D (4.73%), β-caryophyllene (3.02%), γ-muurolene (2.75%), and α-bourbonene (2.27%) were the predominating compounds (Table 1, Figure 1B). Among the identified components, oxygenated monoterpenes (42.66%) are the most abundant, followed by monoterpene hydrocarbons (30.52%), and finally sesquiterpene hydrocarbons (16.82%). The chemical composition of M. spicata volatile oil has been well–documented in many regions, but not in Madinah. In agreement with our findings, previous studies have shown that carvone, limonene, 1,8-cineole, β-caryophyllene, and germacrene D are the main components of the M. spicata volatile oil [11,14,15,16]. Due to differences in ecological parameters, growing location, agronomical practices, as well as environmental conditions, significant variations are expected between M. spicata oil from Madinah and those from other regions, studied by the above researchers. Interestingly, the volatile oil from Madinah appears similar to the Egyptian one. However, they are richer in limonene and oxygenated sesquiterpenes and poor in 1,8-cineol [11].

The oil composition of O. majorana included terpin-4-ol (42.47%), trans-sabinene hydrate (8.52%), γ-terpinene (7.90%), α-terpineol (7.38%), linalool (6.35%), α-terpinene (5.42%), and cis-sabinene hydrate (3.14%) as the main constituents (Table 1, Figure 1C). cis-Sabinene hydrate may be responsible for the aroma properties of the O. majorana plant since it is characterized by intense, spicy, and marjoram aroma [17]. The detected concentration of terpin-4-ol was higher than the one found in the literature [18,19,20] possibly due to rearrangements occurring during the distillation process [21]. The volatile oil of Madinah O. majorana displayed a chemotype similar to Central and Eastern Europe, with a relatively high concentration of linalool and the absence of sabinene, which may positively affect the biological activity of the oil [18]. In addition to a remarkable quantitative variation, p-cymene and sabinene were found in a higher concentration in China, India, Brazil, Egypt, and Iran [17,19,20,22,23]. Again, geographic, climatic, and agronomical aspects are responsible for such variations in oil composition among different cultivation regions [1].

### 2.2. Antioxidant Activity of Volatile Oils

Oxidative damage is the main cause of many diseases, such as cancer, thus, the application of antioxidant activity tests has become one of the most commonly studied topics [1]. The antioxidant capacity of oils from M. longifolia, M. spicata, and O. majorana cultivated in Madinah was evaluated through DPPH radical scavenging activity, ABTS, radical cation decolorization, and β-carotene–linoleate bleaching assays, using TBHQ as a positive control (Table 2). The volatile oils displayed an apparent antioxidant capacity [11,12,24], with lower half-maximal inhibitory activity (IC_50_) values indicating higher activity. O. majorana oil exhibited the highest scavenging ability for DPPH (IC_50_ 1.48 mg/mL), followed by M. spicata (IC_50_ 3.80 mg/mL) and M. longifolia (IC_50_ 6.68 mg/mL). All the tested samples showed lower DPPH radical scavenging activity when compared with the standard (TBHQ). The IC_50_ values of oils under investigation are high compared with the previous reports of Hajlaoui et al. [8], Meloni et al. [24], and Tafrihi et al. [4], which indicate lower activity. In other studies, the antioxidant activity of the oils extracted from M. spicata and M. longifolia cultivated in Algeria and Turkey was relatively low, compared with our results [15,25]. The different antioxidant activities observed, may be ascribed to different chemical constituents and localities geographic [1]. Table 2 shows a similar trend for radical cation (ABTS^•+^) decolorization, with IC_50_ 21.32, 66.80, and 87.30 µg/mL for O. majorana, M. spicata, and M. longifolia oils, respectively. In addition, O. majorana oil had a higher inhibitory effect against the oxidation of linoleic acid and the subsequent bleaching of β-carotene (IC_50_ 10.32 mg/mL), in comparison to the oils of M. spicata (IC_50_ 31.18 mg/mL) and M. longifolia (IC_50_ 54.30 mg/mL).

In line with the above findings, Elansary and Ashmawy [11] compared the activity and composition of four mint types and found that the total antioxidant activity of M. spicata, M. piperita, M. longifolia, and M. suaveolens was respectively 79, 31, 21, 6%, while the results of β-carotene assay for the same species were 85, 55, 45, 20%. A quite lower level of linoleic acid inhibition of M. longifolia oil (36%) has been reported by Gulluce et al. [25], compared to the positive control BHT (96%), in contrast to M. spicata oil (71.2%) [26]. Carvone has been found to be responsible for antioxidant activity in M. spicata oil, with IC_50_ 16.7 μg/mL for the pure terpenoid compound [26]. Further, the presence of limonene as a major component in M. spicata (27.11%, Table 1) and its combination with caryophyllene (3.02%, Table 1) may enhance the antioxidant activity of this oil. Limonene exhibited a higher antioxidant activity (98.74%) compared to thymol (98.57%), linalool (75.88%), and ascorbic acid (62.43%) [27]. The volatile oil of M. longifolia showed a low free radical scavenging activity due to the high content of pulegone (38.42%, Table 1), which has been associated with low reactivity, as reported by Torres-Martíne et al. [27]. In agreement with the present work, pulegone–rich essential oils, such as M. pulegium and M. suaveolens displayed a low free radical scavenging activity [28].

In agreement with the higher total phenolic content and with the presence of active compoundslike linalool (6.35%, Table 1), O. majorana oil showed the highest antioxidant activity (Table 2). Linalool is a well–known phenolic compound with antioxidant activity (IC_50_ 0.61 μg/mL) significantly higher than that of synthetic antioxidants and many essential oils or extracts [29]. In addition, the major component in O. majorana oil is terpin-4-ol (42.47%, Table 1) that exhibits antioxidant activity as reported in tea tree oil by Souza et al. [30]. Mentha species are rich in polyphenols, such as rosmarinic acid, salvianolic acids, hydroxybenzoic acids, caffeoylquinic acids, hydroxycinnamic acids, flavanones, and flavones in M. spicata; rosmarinic acid, salvianolic acid L, dedihydro-salvianolic acid, luteolin-glucuronide, luteolin-diglucuronide, luteolin-glucopyranosyl-rhamnopyranoside, and eriodictyol-glucopyranosyl-rhamnopyranoside in M. longifolia [5]. Similarly, Hossain et al. [31] separated and identified many polyphenols in O. majorana extracts, especially rosmarinic acid by LC–ESI-MS/MS. According to many studies, in addition to the active volatile constituents, antioxidant activity has been correlated to the presence of polyphenols [5].

### 2.3. Evaluation of Molecular Docking

A docking study of the major identified volatiles was conducted on NO to identify the best activity through interaction with the enzyme active sites. The co-crystallized ligands were flexibly re-docked to verify the docking protocol using the MMFF94 force field. The intermolecular interactions between the ligand and the target receptor were evaluated. Validation for the ideal pose was performed by alignment of the X-ray bioactive conformer, with the best-fitted pose of the same compound for the NO. The ideal pose of each molecule was selected according to the energy score and the best fitting to the active site with the least RMSD score.

The binding free energies (∆G) for the volatiles docked at NO are shown in Figure 2, revealing the best poses obtained in the molecular docking analyses. The larger the peaks, the lower the ∆G and consequently the more significant the interaction between the receptor and the ligands with antioxidant ability. cis-verbenyl acetate identified in M. longifolia displayed the best binding affinity compared to other ligands or the control (DEX), with high docking scores (−7.4 kcal/mol), followed by β-elemene (−6.9 kcal/mol) and linalool (−6.8 kcal/mol) detected in M. spicata, and O. majorana, respectively.

Figure 3a–d show the interaction of cis-verbenyl acetate, β-elemene, linalool, and control (DEX) with NO receptor. The higher binding affinity of cis-verbenyl acetate is attributed to the conventional hydrogen bonds formed with TYR186 and LYS187, Pi-alkyl interaction with HIS181, and finally, alkyl interaction with ILE243 and LYS187 (Figure 3a). The number of Pi-sigma interactions (Pi-alkyl) which largely involves charge transfer helps in firmly binding linalool and β-elemene with the receptor residues TYR188, TYR159, and PHE245. Similar alkyl interactions with ILE297, ILE160, PRO298, and LEU299 are shown for both β-elemene and linalool, but the latter has a unique conventional hydrogen bonding with TYR159 and carbon-hydrogen bonding with GLY158 (Figure 3b,c). The similarity of interactions and types of bonds were the main reasons for the close binding affinity and free energy between each ligand and the control (DEX), indicating a good antioxidant ability of the tested molecules. According to the literature, other ligands with different chemical structures have shown an antioxidant activity due to the similarity of interactions and active site with the current study [10,32]. For example, similar to the interacted residues of Figure 3, Costa et al. [10] showed that ASP179, TYR188, ILE243, and HIS181 residues of NO receptor interact with all caffeine analogs used as ligands. In another study, Lountos et al. [32] have shown an interaction between TYR188, ILE160, ILE243, and ASP179 residues and 5-ADP, consistently with our findings.

It is well–known that bioactive compounds containing a phenolic ring and hydroxyl group exhibit a higher antioxidant activity [29]. However, the results of the present molecular docking study open prospects for the use of non-oxygenated and non-phenolic terpenes as efficient antioxidants by reducing oxidative stress. In vivo studies will be necessary to reveal the behavior of different non-oxygenated terpenes as antioxidants since in vitro techniques like DPPH, FRAP, and others are not adequate to show the activity of such categories in reducing oxidative stress.

## 3. Materials and Methods

### 3.1. Plant Materials

*Mentha longifolia* L. (Madany or Habak), *Mentha spicata* L. (Balady), and *Origanum majorana* L. (Doosh) were collected from Abyar Ali, Madinah, Saudi Arabia, in March 2018. Plant samples were identified by a taxonomist at the Department of Botany, Faculty of Science, Taif University and deposited in Madinah Region Municipality-Quality Agency-Food and Environment Laboratory, Madinah, Saudi Arabia with voucher specimens’ numbers AF-36004261–2018, AF-36004262-2018, and AF-36004263–2018.

### 3.2. Chemicals

Diethyl ether and methanol were purchased from Fisher Chemicals (Pittsburgh, USA). The mixture of n-alkanes C_6_–C_26_, authentic compounds, sodium bicarbonates, linoleic acid ( ≥ 99%), Tween 40, β-carotene ( ≥ 97%), Folin-Ciocalteu reagent for total phenolics, 2,2′-diphenyl-1-picrylhydrazyl (DPPH),2,2′—azino-bis-3-ethylbenzthiazoline-6-sulphonic acid (ABTS, purity >99%), methanol, TBHQ (tert-Butyl hydroquinone), and gallic acid were obtained from Sigma Aldrich Chemical Co. (St. Louis, MO, USA).

### 3.3. Extraction of the Volatile Components

Air-dried aerial parts of *M. longifolia*, *M. spicata*, and *O. majorana* (100 g) in three replicates that were cut into small pieces and subjected to hydrodistillation for 3 h, using a Clevenger type apparatus. The volatile-compounds extract was dried using anhydrous sodium sulfate, stored in airtight glass vials covered with aluminum foil at (−20 °C) until analysis [1].

### 3.4. Gas Chromatography-Mass Spectrometry (GC–MS)

The components of the essential oil obtained using hydrodistillation were analyzed using a GC–MS apparatus. The separation was performed on a Trace GC Ultra Chromatography system (Thermo Scientific, USA) equipped with an ISQ-mass spectrometer (Thermo Scientific, USA) with a 60 m × 0.25 mm × 0.25 μm-thick TG-5MS capillary column (Thermo Scientific, USA). The column separation was programmed from 50 °C with a holding time of 3 min, and then the temperature was increased at a rate of 4 °C per min to 140 °C with a holding time of 5 min. After that, the temperature was increased at 6 °C per minute to 260 °C for a 5-min isothermal holding time. The injector temperature was 180 °C, the ion source temperature was 200 °C, and the transition line temperature was 250 °C. The carrier gas was helium with a constant flow rate of 1.0 mL/min. The mass spectrometer had a scan range from m/z 40–450, and the ionization energy was set at 70 eV. The identification of compounds was based on matching with the MS computer library (NIST library, 2005 version) and compared with those of authentic compounds and published data [33]. The relative percentage of the oil constituents was calculated from the GC peak areas. Kovat’s index was calculated for each compound, using the retention times of a homologous series of C_6_–C_26_ n-alkanes and by matching with the literature [4,7,11,15,16,19,22,23].

### 3.5. Antioxidant Activity Measurements

DPPH radical scavenging assay: Potential antioxidant activity of the oils under investigation was assessed according to the methods reported by Hatano et al. [34] in comparison to a synthetic antioxidant used in the food industry, tert-butyl hydroquinone (TBHQ). Serial of essential oils were applied with concentrations of 0.5–10 mg/mL of methanol. After incubation with DPPH solution, the absorbance was measured at 517 nm using a spectrophotometer (Evolution 300 Thermo UV-VIS); all tests were run in three replicates, and the results were averaged. A positive control (TBHQ) at concentrations ranging from 0.1–5 mg/mL was applied in parallel. The radical scavenging activity of extracts was expressed as the IC50 value (mg/mL), i.e., the concentration necessary to decrease the DPPH concentration by 50%.

β-Carotene bleaching assay: The antioxidant activity of the oils was determined using a β-carotene/linoleic acid system, as described by Taga et al. [35] in comparison to TBHQ. Different concentrations of oils ranging from 1-100 mg/mL were applied to the β-carotene/linoleic acid system, where the absorbance was measured at 470 nm over a 60-min period. All determinations were performed in triplicate.

The ABTS free radical–scavenging assay: The antioxidant activity of the oils under investigation was determined by their ability to decolorize the radical cation (ABTS^•+^) as previously detailed [36]. The method comprises the addition of 180 µL of ABTS solution to 20 µL of the sample solution in methanol at different concentrations (1–100 µg/mL). After 10 min, the percentage inhibition at 734 nm was calculated for each concentration relative to a blank absorbance (methanol). The scavenging capability of ABTS was calculated using the following equation: ABTS^•+^ scavenging activity (%) = ((A_Control_ − A_Sample_)/A_Control_) × 100.

Total phenolic content: Total phenolic content of the essential oils was determined using the Folin-Ciocalteu reagent according to a method modified from Singleton et al. [37] using gallic acid as the standard. The reaction mixtures were incubated in a thermostat at 45 °C for 45 min before the absorbance at 765 nm was measured. The total phenolic contents were calculated based on the calibration curve of gallic acid and expressed as gallic acid equivalents (GAE), in milligrams per gram of the sample.

### 3.6. Molecular Docking

A flexible molecular docking study was carried out on NADPH oxidase (NO) responsible for the production of ROS using the Swissdock server, accessed on 12 May 2021 (www.swissdock.ch/docking#) based on grid center coordinates and size described by Costa et al. [10]. NO-Protein was obtained from the Protein Data Bank (PDB) with code (2CDU) and prepared as a receptor by removing waters and co-crystallized ligands (ADP and FAD) then protonated using Pymol software [38]. Meanwhile, the 3d structure of ligands downloaded from the PUBCHEM database, accessed on 12 May 2021 (http://pubchem.ncbi.nlm.nih.gov/) were optimized using the MMFF94 force field by ChemDraw Software (Ver. 18.0) [39]. Docking results(∆G) were viewed using Chimera 1.14 software, while profiles of interaction and visualization were performed using Discovery Studio software (Ver. 20.1.0.19295) [40].

### 3.7. Statistical Analysis

Statistical analyses were performed using SPSS software version 16. The data were expressed as mean ± SD and analyzed using the student’s t-test and variance analysis.

## 4. Conclusions

In the present study, volatiles of dried *Mentha longifolia* L., *Mentha spicata* L., and *Origanum majorana* L. cultivated in Madinah Monawara, KSA, were extracted using hydrodistillation followed by separation and identification using GC–MS. Pulegone and 1,8-cineole were the major components in *M. longifolia* oil, while carvone and limonene predominate in *M. spicata* oil and terpin-4-ol in *O. majorana* oil. Madinah *O. majorana* oil was associated with a higher scavenging ability and a more significant inhibiting effect toward the oxidation of linoleic acid compared to *M. spicata* and *M. longifolia*, due to the presence of many volatiles with phenolic nature and, therefore, antioxidant activity as well as higher total phenolic content. A molecular docking study revealed that non-oxygenated terpenes display higher activity in reducing oxidative stress through interaction with NADPH oxidase residues. In vivo studies will be necessary to reveal the behavior of different non-oxygenated terpenes as antioxidants since the in vitro techniques are not adequate to show the activity of such categories in reducing oxidative stress.

## Figures and Tables

**Figure 1 molecules-26-04145-f001:**
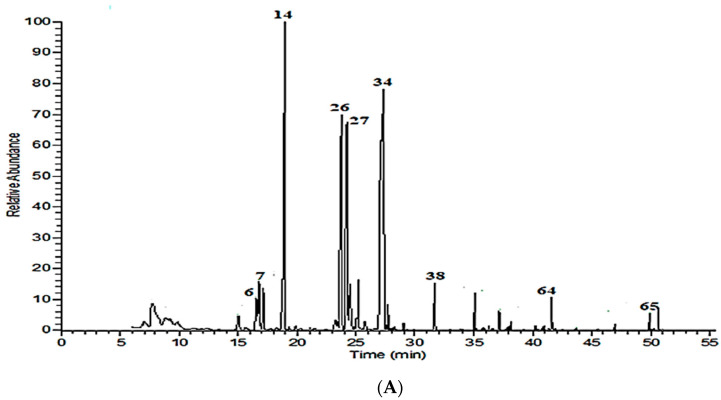
Volatile chromatograms for Madinah, Saudi Arabia (**A**) *M. longifolia*, (**B**) *M. spicata*, and (**C**) *O. majorana*.

**Figure 2 molecules-26-04145-f002:**
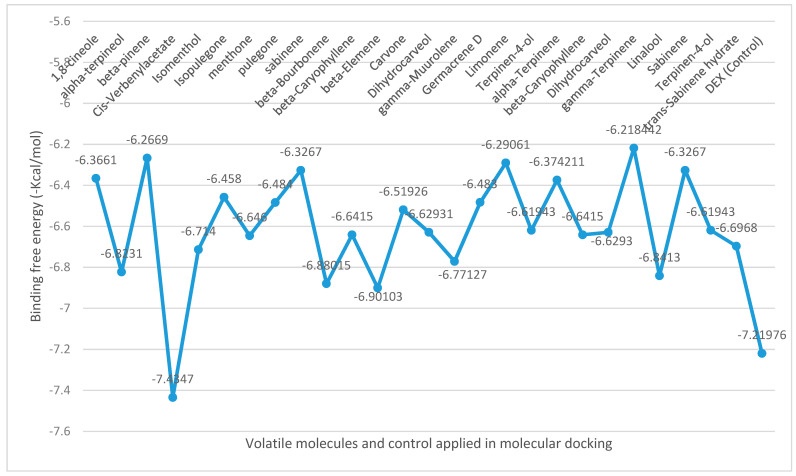
Binding free energy values calculated through the molecular docking of the major volatiles identified in lamiaceous plants and NO receptor.

**Figure 3 molecules-26-04145-f003:**
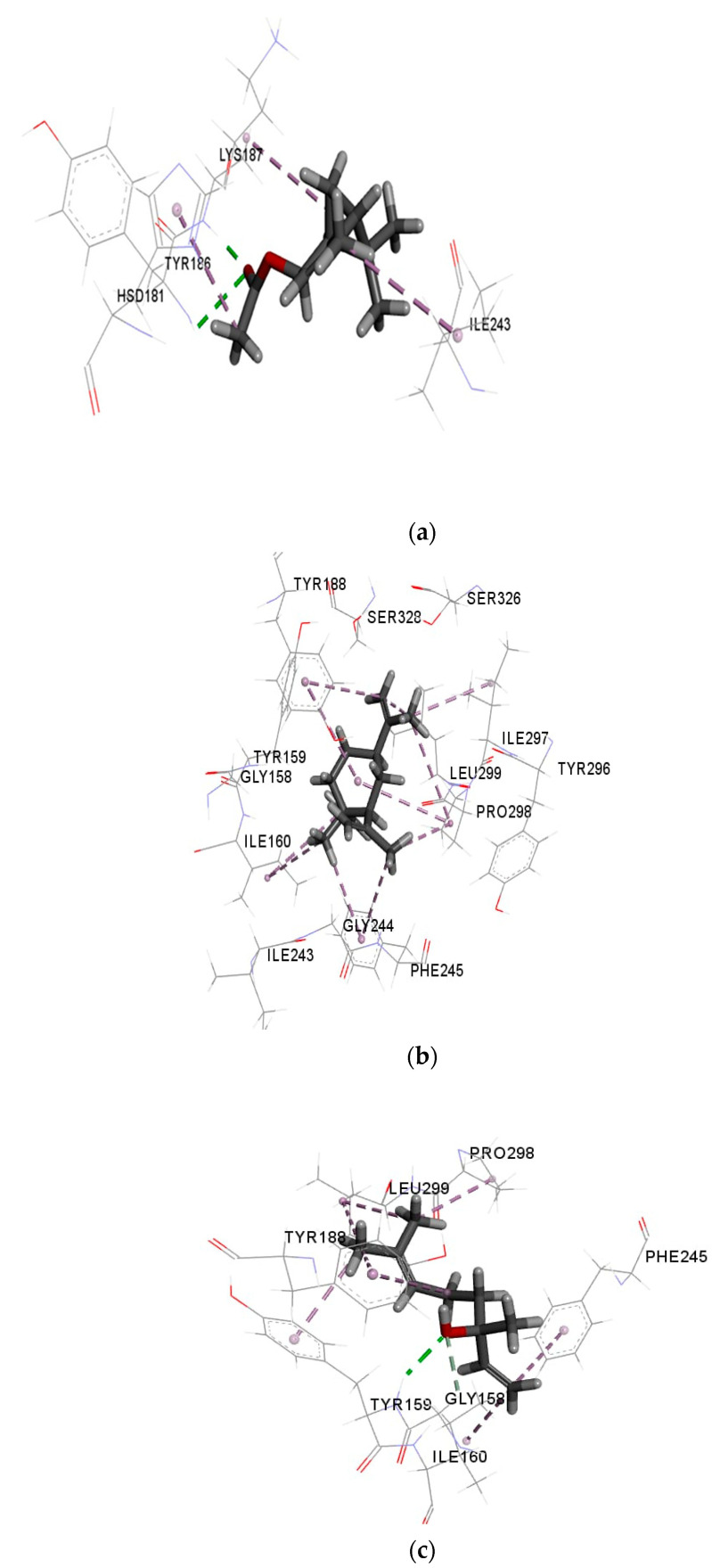
Interactions of (**a**) cis-verbenyl acetate, (**b**) β-elemene, (**c**) linalool, and (**d**) DEX (control) with NO receptor.

**Table 1 molecules-26-04145-t001:** Identification of the volatile constituents of *M. longifolia*, *M. spicata*, and *O. majorana* oils using GC-MS.

S/N	Compound	RI ^a^	LRI ^b^	Area%			Identification Method ^c^
				*M. longfolia*	*M. spicata*	*O. majorana*	
1	Methyl cyclohexane	726	716	0.90 ± 0.05	n.d.	n.d.	RI, MS
2	3-Hexen-1-ol	855	851	n.d.	**1.33 ± 0.6**	0.46 ± 0.06	RI, MS
3	α-Thujene	932	931	n.d.	0.51 ± 0.14	0.12 ± 0.04	RI, MS
4	α-Pinene	937	939	0.74	0.65 ± 0.11	0.18 ± 0.07	RI, MS
5	3- Methyl cyclohexanone	955	952	0.31 ± 0.06	n.d.	n.d.	RI, MS
6	Sabinene	978	976	**3.09 ± 0.92**	0.50 ± 0.04	**2.50 ± 1.05**	RI, MS, STD
7	β-Pinene	982	980	**2.25 ± 0.61**	n.d.	n.d.	RI, MS, STD
8	α-Myrcene	986	991	n.d.	0.66 ± 0.07	**1.19 ± 0.73**	RI, MS, STD
9	3- Octanol	992	993	n.d.	1.04 ± 0.08	n.d.	RI, MS
10	α-Phellandrene	1003	1005	n.d.	n.d.	0.34 ± 0.09	RI, MS
11	α-Terpinene	1017	1018	n.d.	0.07 ± 0.01	**5.42 ± 1.42**	RI, MS, STD
12	*p*-Cymene	1025	1026	0.20 ± 0.03	0.11 ± 0.02	0.57 ± 0.1	RI, MS
13	Limonene	1034	1031	0.49 ± 0.06	**27.11 ± 2.6**	**2.57 ± 0.94**	RI, MS, STD
14	1,8- Cineole	1039	1033	**15.60 ± 1.21**	0.61 ± 0.08	0.33 ± 0.05	RI, MS, STD
15	β-Ocimene (*Z*-)	1042	1040	n.d.	0.08 ± 0.02	n.d.	RI, MS
16	β-Ocimene (*E*-)	1052	1050	n.d.	0.14 ± 0.03	n.d.	RI, MS
17	γ-Terpinene	1063	1062	n.d.	0.33 ± 0.08	**7.90 ± 1.45**	RI, MS, STD
18	*cis*- Sabinene hydrate	1070	1068	n.d.	0.18 ± 0.04	**3.14** ± 0.76	RI, MS
19	Terpinolene	1090	1088	n.d.	0.18 ± 0.03	n.d.	RI, MS
20	*trans*-Sabinene hydrate	1098	1097	0.36 ± 0.09	n.d.	**8.52 ± 1.14**	RI, MS
21	Linalool	1101	1098	0.10 ± 0.02	0.10 ± 0.01	**6.35 ± 1.09**	RI, MS, STD
22	*p*-Menth-8-en-1-ol	1142	1140	n.d.	n.d.	**1.24 ± 0.47**	RI, MS
23	Verbenol	1144	1140	0.28 ± 0.08	n.d.	n.d.	RI, MS
24	*cis*-β-Terpineol	1146	1144	n.d.	0.59 ± 0.06	n.d.	RI, MS
25	Isothujanol	1149	1146	0.18 ± 0.04	n.d.	n.d.	RI, MS
26	Menthone	1155	1154	**13.20 ± 2.4**	0.83 ± 0.53	n.d.	RI, MS, STD
27	Isopulegone	1177	1174	**9.81 ± 1.1**	n.d.	n.d.	RI, MS
28	Terpin-4-ol	1180	1177	n.d.	**1.76** ± 0.89	**42.47 ± 3.12**	RI, MS, STD
29	Isomenthol	1183	1182	**1.58 ± 0.8**	n.d.	n.d.	RI, MS
30	α-Terpineol	1191	1189	**1.57 ± 0.9**	n.d.	**7.38 ± 1.03**	RI, MS, STD
31	Dihydrocarveol	1192	1192	n.d.	**1.33** ± 0.93	**1.89 ± 0.73**	RI, MS
32	*cis*-Dihydrocarvone	1195	1193	n.d.	0.6	n.d.	RI, MS
33	Verbenone	1207	1204	0.66 ± 0.1	n.d.	n.d.	RI, MS
34	Pulegone	1240	1237	**38.42 ± 3.9**	n.d.	n.d.	RI, MS, STD
35	Carvone	1244	1242	0.80 ± 0.12	**35.14 ± 3.72**	n.d.	RI, MS, STD
36	Piperitone	1251	1252	0.2	0.27 ± 0.05	n.d.	RI, MS
37	Linalyl acetate	1261	1257	n.d.	n.d.	0.27 ± 0.08	RI, MS
38	*cis*-Verbenyl acetate	1283	1282	**1.75 ± 0.83**	n.d.	n.d.	RI, MS
39	Bornyl acetate	1287	1285	n.d.	0.50 ± 0.18	n.d.	RI, MS
40	Menthyl acetate	1291	1294	n.d.	0.29 ± 0.02	n.d.	RI, MS
41	δ-Terpinyl acetate	1307	1313	n.d.	n.d.	0.16 ± 0.02	RI, MS
42	Dihydrocavyl acetate (iso-)	1338	1325	n.d.	0.12 ± 0.07	n.d.	RI, MS
43	*trans*-Carvyl acetate	1347	1337	n.d.	0.12 ± 0.08	n.d.	RI, MS
44	δ-Elemene	1350	1339	n.d.	n.d.	0.12 ± 0.01	RI, MS
45	*cis*-Carvyl acetate	1363	1362	n.d.	0.40 ± 0.06	n.d.	RI, MS
46	Neryl acetate	1369	1365	n.d.	n.d.	0.86 ± 0.16	RI, MS
47	α-Copaene	1367	1376	n.d.	0.18 ± 0.06	0.22 ± 0.04	RI, MS
48	β-Bourbonene	1376	1384	n.d.	**2.27 ± 0.79**	n.d.	RI, MS
49	Geranyl acetate	1381	1383	n.d.	n.d.	0.27 ± 0.06	RI, MS
50	β-Elemene	1392	1391	n.d.	**1.75 ± 0.45**	n.d.	RI, MS
51	*cis*-Jasmone	1400	1394	n.d.	0.49 ± 0.17	n.d.	RI, MS
52	β-Caryophyellene	1428	1418	0.97 ± 0.21	**3.02 ± 1.05**	**1.77 ± 0.54**	RI, MS, STD
53	γ-Elemene	1430	1433	n.d.	n.d.	**1.67 ± 0.66**	RI, MS
54	α-Humulene	1450	1454	0.13 ± 0.06	0.38 ± 0.02	n.d.	RI, MS
55	Muurola-4(14),5-diene (*cis**-*)	1464	1460	0.13 ± 0.08	n.d.	n.d.	RI, MS
56	γ-Muurolene	1478	1477	n.d.	**2.75 ± 0.99**	n.d.	RI, MS
57	Germacrene D	1482	1480	0.42 ± 0.06	**4.73 ± 1.43**	n.d.	RI, MS
58	Bicyclogermacrene	1496	1494	n.d.	0.54 ± 0.34	0.09 ± 0.02	RI, MS
59	α-Muurolene	1502	1499	0.16 ± 0.05	0.84 ± 0.26	n.d.	RI, MS
60	δ-Cadinene	1515	1524	0.22 ± 0.11	0.36 ± 0.06	n.d.	RI, MS
61	Spathulenol	1577	1576	n.d.	0.33 ± 0.03	0.26 ± 0.03	RI, MS
62	Caryophellene oxide	1582	1581	n.d.	0.47 ± 0.08	0.16 ± 0.01	RI, MS
63	Cubenol	1641	1642	n.d.	0.96 ± 0.21	n.d.	RI, MS
64	α-Cadinol	1655	1553	0.73 ± 0.08	n.d.	n.d.	RI, MS
65	Phytol	1955	1949	0.66 ± 0.12	n.d.	n.d.	RI, MS
-	Total	-	-	**95.91**	**94.62**	**98.42**	

^a^ RI: retention indices calculated on DB-5 column using alkanes standards; ^b^ LRI: retention indices according to literature; ^c^ Confirmed by comparison with the retention indices, the mass spectrum of the authentic compounds, and the NIST mass spectra library data; n.d: not detected.

**Table 2 molecules-26-04145-t002:** Antioxidant activity of essential oils for *M. longifolia*, *M. spicata*, and *O. majorana* cultivated in Madinah in comparison to the synthetic antioxidant TBHQ.

Volatile Oil	IC_50_ (mg/mL) (DPPH) ^a^	IC_50_ (µg/mL) ABTS Assay	IC_50_ (mg/mL) β-Carotene Bleaching Assay	Total Phenolic Content ^a^ mg GA/g (for 1mg/mL)
*M. longifolia*	6.68 ± 0.12	87.30 ± 1.32	54.30 ± 0.23	24.10 ± 0.52
*M. spicata*	3.80 ± 0.31	66.80 ± 1.16	31.18 ± 0.51	42.30 ± 0.43
*O. majorana*	1.48 ± 0.08	21.32 ± 0.72	10.32 ± 0.41	79.10 ± 1.01
(TBHQ)	1.01 ± 0.05	12.33 ± 0.32	4.12 ± 0.11	--

^a^ Values represent averages ± standard deviations for triplicate experiments.

## Data Availability

The data presented in this study are available in this article.

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
