# Peer review of "Antioxidant Activity and Molecular Docking Study of Volatile Constituents from Different Aromatic Lamiaceous Plants Cultivated in Madinah Monawara, Saudi Arabia"

_molecules, 2021, doi:10.3390/molecules26144145_

Round 1

Reviewer 1 Report

In the original paper: "Antioxidant activity and molecular docking study of volatile constituents from different aromatic lamiaceous plants cultivated in Madinah Monawara, Saudi Arabia," the Authors present an analysis of the composition of essential oils obtained from three species of plants in Saudi Arabia. In my opinion, the work still requires many improvements before its publication. Both the content and the methodology require elaboration. I also think that the authors should propose more tests to study the antioxidant properties of the tested substances (many of them are neither expensive nor difficult). Linguistic correctness also raises doubts. Below I present my suggestions, which will help to improve this manuscript.

Abstract

- The construction of the abstract must be rethought by the Authors. It lacks a clearly defined purpose. In addition, information on the antioxidant activity was presented somewhat vaguely - without specifying the method using which the results were obtained. The GC-MS data is too accurate. The conclusion is faintly marked. The abstract needs to be corrected.

Keywords

- Keywords should be different from the words used in the title - it makes it easier to notice the article.

Introduction

- I think the Authors should work at the English used in the manuscript. Sometimes the sentences are not clear, which makes it difficult to understand well the content presented.

- In my opinion the first paragraph of Introduction is unnecessarily developed, geographic coordinates are redundant - a general description would suffice.

- The short excerpt on antioxidants adds little.

- the Authors could include more information on the biological activity of all studied species.

- There is an abbreviation "DEX" that has not been explained in the Introduction.

Methodology

- I would suggest that you separate information about plant material and chemicals.

- The extraction description should be much more detailed (there is a lack of data on the weight of raw materials taken for extraction and the volume of the solvent). There is also no information on the amount/volume of the obtained volatile-compounds extracts.

- Please indicate the modifications that applied to the method of determining the Total phenolic content.

- The methodology of antioxidant testing lacks information on how the results of the study are expressed.

- Information on the concentrations at which the trials were conducted is missing.

The main part of the manuscript

- Editorial errors appear, such as lack of italics in the names of Latin plants, unnecessary italics in the caption Figure 1, lowercase letters instead of capital letters (m. Longifolia), etc.

- Authors should improve some Figures in quality.

- The tables should also be more graphically polished.

- It is unclear what precisely refers to the unit "mg/ml" used to describe the IC50 parameter in the DPPH method. It does not follow from the methodology that a solid substance was weighed? So what does "mg" refer to, and what does "ml" refer to?

- The expressed activity values ​​should be corrected in such a way that there are one or two digits after the decimal point (it is about unifying the defined values).

- This sentence sounds like linalool was treated as a phenolic compound: “In agreement with the higher total phenolic content detected and effective active constituents like linalool (6.35%, Table 1) ...”

- What relationship has the potential sedative or anesthetic effects of terpinen-4-ol to this antioxidant effect - this inclusion is not necessary.

- I do not see fragments constituting a discussion about the research on the content of polyphenols.

Conclusion

Are the phrase accurate: “Madinah O. majorana oil showed a higher scavenging ability and a more significant inhibiting effect toward the oxidation of linoleic acid than M. spicata and M. longifolia cause of the presence of many volatiles with phenolic nature and, therefore, antioxidant activity as well as higher total phenolic content.", as a concern of phenolic structure in volatile compounds from examined substances.

Reviewer 2 Report

The study by the authors attempted to comparatively investigate the volatile constituents of Mentha longifolia L., Mentha spicata L., and Origanum majorana L., while reporting their antioxidant and phenolic contents. Judging by the IC50 value for the DPPH assay, O. majorana's oil seems to potentiate the highest antioxidant cabability with M. longifolia being the least and this was correlated with the total phenolic content of the respective botanicals.

While the concept of the work seems sound, the followings need to be addressed:

  1. For the sake of reproducibility, kindly provide detailed information on how the IC50 and % for DPPH and bleaching assays, respectively, were determined. Similar information must also be included for the total phenolic acid content evaluation.
  2. NADPH is involve in the production of superoxide from oxygen and NADPH and not general ROS as erroneously stated by the authors in the manuscript. This must be corrected in the entire manuscript and provide justification for its use as target for oxidative stress in this study knowing fully well that there are other druggable targets of REDOX importance.
  3. Please, provide details of how the ligands (volatile oils) were optimized prior to docking and the version of Discovery Studio used.
  4. It was stated that the plants used were identified by a Taxonomist without providing the voucher details of each species. This must be included in the manuscript.
  5. Provide pictures with better resolutions for the three chromatograms presented (Figure 1) and clearly label the axes in Figure 2 with the unit of binding free energy also included.
  6. Table 2 is confusing. Why two representations for DPPH? Clarify the ambiguity.
  7.  Page 8, Line 207 - 212: Be explicit on the specific ligand with respect to comparison being made regarding the compounds in Figure 3. Also, explain and justify the contribution(s) of each bond type observed towards the stability of the enzyme-ligand complex and not just mere mention of the amino acid residues.
  8. The entire manuscript need grammatical overhauling and improvement for syntax errors. For instance, there should be 'is' between Madinah and located (Page 2, Line 38). Page 2, Lines 59-62 is not clear. These and several others need to be corrected in the entire manuscript.
  9. While the CONCLUSION lacks recommendation(s) for further studies and significance of the study, the ABSTRACT is devoid of both 'conclusion' and 'future prospects'. These are key components of a good scientific report.  

Reviewer 3 Report

Antioxidant assays by DPPH and B-carotene methods alone are very preliminary, needing many further assays. 
Moreover, the activity of these terpenes reported by the manuscript has already been widely known for many years. No new information that could positively impact the field of research is provided. 
The fact that a compound exhibits antioxidant activity does not mean that this compound will be able to inhibit NADPH oxidase.

There are no MD simulation studies or binding calculation studies are performed, there is no energy calculations by MMPBSA/MMGBSA or similar approaches. Without MD simulation, just on the basis of docking the prediction has limited scope and applications. Thus, it is essential to carryout the MD simulation studies.

As a suggestion to the authors, I suggest that they carry out further tests (FRAP and ABTS) and also perform synergism or formulation tests (i.e. nanocapsules) of these compounds.

Round 2

Reviewer 1 Report

The authors significantly revised the article, taking into account the comments of the reviewers. In this form, the article is of considerable interest and can be published.

Author Response

Thank you for your kind reply.

Reviewer 2 Report

The manuscript by Farouk et al. has been revised with most of the suggestions and recommendations effected. However, the followings still need to be addressed:

  1. For the molecular docking, it was suggested that authors should detail how the ligands (oil constituents) were optimized but this has not been addressed. Rather, authors only mentioned 'optimization' without explaining what was done for the ligands to be optimized. This must be revised accordingly.
  2. Similarly, sufficient details have not been provided on how the IC50 values were determined. Again, they were only defining IC50 instead of detailing how it was actually evaluated.
  3.  Despite that it was claimed that the manuscript has been edited by a professional house “Best Edit Proof”, the entire manuscript still needs grammatical overhauling and improvement for syntax errors.

Author Response

Thank you for your kind suggestions. Please see the attachment.

Reviewer 3 Report

letter to editor

Author Response

Thanks for your kind suggestions.